# IBF-R Regulates IRE1α Post-Translational Modifications and ER Stress in High-Fat Diet-Induced Obese Mice

**DOI:** 10.3390/nu14010217

**Published:** 2022-01-04

**Authors:** Hwa-Young Lee, Geum-Hwa Lee, Young Yoon, The-Hiep Hoang, Han-Jung Chae

**Affiliations:** 1Department of Pharmacology, Institute of New Drug Development, Medical School, Jeonbuk National University, Jeonju 54896, Jeollabuk-do, Korea; youngat84@gmail.com; 2Non-Clinical Evaluation Center, Biomedical Research Institute, Jeonbuk National University Hospital, Jeonju 54907, Jeollabuk-do, Korea; heloin@jbnu.ac.kr (G.-H.L.); drhiep.ydhue@gmail.com (T.-H.H.); 3Imsil Cheese & Food Research Institute, Doin 2-gil, Seongsu-myeon, Imsil-gun 55918, Jeollabuk-do, Korea; kuburi79@icf.re.kr; 4Research Institute of Clinical Medicine, Jeonbuk National University-Biomedical Research Institute, Jeonbuk National University Hospital, Jeonju 54907, Jeollabuk-do, Korea; 5School of Pharmacy, Jeonbuk National University, Jeonju 54896, Jeollabuk-do, Korea

**Keywords:** obesity, *Rhus verniciflua*, high-fat diet, S-nitrosylation, ER stress, adipogenesis

## Abstract

Obesity is a global health issue linked to the heightened risk of several chronic diseases. *Rhus verniciflua* (RV) is a traditional food supplement used for a range of pharmacological effects such as antitumor, antioxidant, α-glucosidase inhibitory effects, hepatitis, and arthritis. Despite the traditional medicinal values, scientific evidence for its application in obesity is inadequate and unclear. Thus, this investigation was designed to evaluate the anti-obesity effects of IBF-R, an RV extract, using a high-fat diet (HFD) model. The study has six groups: chow diet group; chow diet with 80 mg/kg IBF-R; HFD group; IBF-R group with 20, 40, and 80 mg/kg. IBF-R supplementation significantly regulated the weight gain than the HFD fed mice. Further, IBF-R supplementation lowered the expressions of adipogenic transcription factors such as SREBP-1c, C/EBPα, FAS, and PPAR-γ in white adipose tissue (WAT) of diet-induced obese mice. In addition, IBF-R supplementation reduced the lipogenic gene expression while enhancing genes was related to fatty acid oxidation. Obesity is linked to redox-based post-translational modifications (PTMs) of IRE1α such as S-nitrosylation, endoplasmic reticulum (ER) stress, and chronic metabolic inflammation. The administration of IBF-R inhibits these PTMs. Notably, IBF-R administration significantly enhanced the expression of AMPK and sirtuin 1 in WAT of HFD-fed mice. Together, these findings reveal the IRE1α S-nitrosylation-inflammation axis as a novel mechanism behind the positive implications of IBF-R on obesity. In addition, it lays a firm foundation for the development of *Rhus verniciflua* extract as a functional ingredient in the food and pharmaceutical industries.

## 1. Introduction

Obesity is abnormal fat accumulation in adipose tissues and characterized by body mass index (BMI) higher than 30 kg/m^2^ and is associated with multiple chronic diseases such as type 2 diabetes, cardiovascular disorders, and different types of cancers [1]. White adipose tissue (WAT) executes a variety of metabolic tasks under physiological conditions. However, WAT loses its functional capabilities with obesity negatively affecting its storage of surplus energy. This process leads to ectopic fat accumulation in several tissues influencing metabolic homeostasis [2]. Thus, developing strategies against lipogenesis and lipid dysmetabolism in obesity and associated diseases has potential therapeutic benefits [3]. Furthermore, previous investigations demonstrate that visceral adipose tissue has higher inflammatory and immunological cells than subcutaneous adipose tissue. Hence, regulating visceral adipose tissue development by lowering lipogenesis and enhancing lipid oxidation could be a promising approach to treat obesity and associated metabolic dysfunction. 

Obesity is a complicated health condition involving multiple molecular mechanisms. Aging impairs endoplasmic reticulum (ER) chaperones and foldases, affecting protein folding capabilities [4]. In addition, redox signaling imbalances or abnormal redox-associated post-translational modifications (PTMs) alter protein structure or function, resulting in pathogenic diseases. However, the underlying mechanisms underpinning obesity-related redox imbalance, and ER proteostasis dysfunction, as well as their significance in disease progression, remain unknown. Obese adipose tissues are prone to stressful conditions such as hypoxia, ER stress, and oxidant stress [5,6]. Recent reports suggest up-regulation of ER stress markers such as phosphorylated PERK (p-PERK), phosphorylated α-subunit of eukaryotic translational initiating factor 2 (eIF2α), and BiP in liver and adipose tissues of high-fat-diet (HFD) mice [6]. Moreover, multiple investigations have found enhanced free fatty acids (FFAs) during obesity, potentially inducing ER stress in adipocytes and other cells [7,8]. However, underlying mechanisms that induce ER stress in adipose tissue or the relationship between ER stress and chronic inflammation during obesity are unclear. Here, we discovered that excessive reactive nitrogen species (RNS) build up during obesity, causing nitro-oxidative stress and triggering cysteine oxidation. These processes induce alterations in ER membrane protein, IRE1α. Further, the functioning of IRE1α is influenced by S-nitrosylation of IRE1α. 

*Rhus verniciflua* (RV) is a traditional food supplement used for its medicinal values for centuries in Eastern Asia. The use of RV was demonstrated to have several pharmacological effects, including α-glucosidase inhibitory effects, antioxidant, antitumor, anti-inflammatory, and antibacterial effects [9,10,11]. The abundance of flavonoids and polyphenols such as fisetin, sulfuretin, quercetin, fustin, and butein in RV are thought to be responsible for these pharmacological actions. Additionally, modifications in hepatic metabolism and related dysfunctions are strongly linked to RV [12,13]. Still, scientific evidence for its beneficial effects on obesity is inadequate and unclear. Thus, we hypothesized that RV could protect against obesity by modulating obesity-related instability in the ER function. In this study, IBF-R, RV extract is extracted through water extraction as it is more popular than other methods. Here, efforts are put to investigate the potential impact and mechanism behind the beneficial effects of IBF-R on obesity. 

## 2. Materials and Methods

### 2.1. The Preparation of IBF-R

IBF-R, RV extract was extracted as described previously [14]. Finely powdered RV was hot air dried, boiled, condensed, and lyophilized to obtain dried RV extract (IBF-R). The yield of IBF-R was close to 4%. The quality control was performed using standard compounds such as Fisetin as described before [15].

### 2.2. Animal Studies

Male C57BL/6J mice (6 weeks old) were procured from Orient Science Co. (Seongnam, Korea). All the animals were kept in well-ventilated cages at 22 ± 2 °C under a standard light-dark cycle. All the mice used in this study were acclimatized for a week prior to use. Later, 7 weeks old mice were randomly selected and separated into six groups, with 8 mice in each group. NCD mice receiving water were assigned as Group 1. NCD mice receiving 80 mg/kg of IBF-R were assigned as Group 2. HFD-fed mice receiving water were assigned as Group 3. Group 4 received 20 mg/kg IBF-R, Group 5 received 40 mg/kg IBF-R and Group 6 received 80 mg/kg IBF-R. The experimental diet was continued for up to 12 weeks. At the end of the experiment, all the animals were sacrificed to collect samples. Collected whole blood was stored immediately at 2 °C for 30 min and centrifuged to separate the serum. All the samples were maintained at −80 °C until use. The animal tests were performed by following the Jeonbuk National University hospital’s animal care and use committee guidelines (CUH-IACUC-2019-10).

### 2.3. Biochemical Analysis

All the biochemical tests were performed using commercial kits. Whole blood was collected using EDTA vacutainers. Serum was separated by centrifuging the blood at 3000× *g* 10 min at 4 °C. Triglyceride (TG, AM1575K) and total cholesterol (TC, AM202K) were detected using commercial test kits from Asan Pharmaceutical, Seoul, Republic of Korea. Plasma levels of adiponectin (CSB-E07272m, CUSABIO, Houston, TX, USA) and leptin (CSB-E04650m, CUSABIO, Houston, TX, USA) were determined using commercial sandwich ELISA kits. The absorbance at 450 nm was detected using Multiskan SkyHigh microplate spectrophotometer (Thermo Fisher Scientific, Inc., Waltham, MA, USA). The unknown sample concentration (ng/mL) was determined from the standard curve.

### 2.4. Immunoblotting 

Immunoblotting was performed as described earlier [15]. All the immunoblots were probed with relevant antibodies. Antibodies used in this study are AMP-activated kinase (AMPK, #2532) and phosphorylation of AMP-activated kinase (p-AMPK, P-2535) (Cell Signaling Technologies, Inc., Danvers, MA, USA) and sterol regulatory element-binding protein (SREBP-1c, sc-36553), peroxisome proliferator-activated receptor γ (PPAR-γ), CCAAT/enhancer-binding protein α (C/EBP1α, sc-166258), fatty acid synthase (FAS, sc-74540), sirtuin 1 (SIRT1, sc-74465), β-actin (sc-47778, Santa Cruz, Biotechnologies, Inc., Santa Cruz, CA, USA). 

### 2.5. Oxyblot Assay 

Tissue carbonylated protein was measured via OxyBlot Protein oxidation Kit (Millipore, Billerica, MA, USA) as per guidelines set by the producer. Briefly, samples were incubated 1:1 in 2,4-dinitrophenylhydrazine (DNPH) and added 2-mercaptoethanol for protein denaturing. Later, a neutralization reagent was used to cease the reaction, and samples were loaded for immunoblotting as described in previous sections. 

### 2.6. Histological Analysis and Immunohistochemistry (IHC)

The liver and WAT were dissected and fixed with 10% formalin. Later, paraffin-embedded tissue blocks were sectioned 4 µm and stained with H&E. Similarly, for IHC, formalin-fixed, paraffin-embedded tissues were processed as described previously [15]. 

### 2.7. Reverse Transcription Polymerase Chain Reaction (RT-PCR)

Total RNA from tissue was isolated with TRIzol (Invitrogen, Carlsbad, CA, USA). Extraction was performed with 1 μg of RNA with oligo dT primers. SYBR premix Ex Taq kit (TaKaRa Bio Inc., Shiga, Kusatsu, Japan) was used for quantitative PCR. Quantification was performed by a comparative cycle threshold (Ct) method, and the resultant product was normalized to β-actin expression. 

### 2.8. Lipid Peroxidation Measurement

Adipose tissue lipid peroxidation was measured with OxiSelect^TM^ TBARS Assay Kit (#STA-330, Cell Biolabs, Inc. San Diego, CA, USA), as per the guidelines set by the manufacturer. Briefly, samples were incubated with lysis buffer and allowed to react with thiobarbituric acid (TBA). Cooled samples were centrifuged to collect the supernatant, and absorbance was measured at 532 nm.

### 2.9. Detection of High Molecular Weight Complex (HMWC)

HMWC’s were detected by following previous reports [16]. Homogenized whole adipose tissues were used to detect HMWC of Protein Disulfide Isomerase (PDI, ADI-SPA-891-F, Enzo Life Sciences, Inc., Farmingdale, NY, USA). Washed adipose tissue supplemented with 1mM phenylmethylsulfonyl fluoride (PMSF) is lysed, homogenized, sonicated, cleared, and quantified. Next, about 50 µg of total protein was separated under non-reducing conditions using 8% polyacrylamide gels.

### 2.10. Detection of S-Nitrosylation 

Detection of S-nitrosylation was performed with Pierce^TM^ S-nitrosylation kit (#90105; Thermo Scientific, Waltham, MA, USA) as per the guidelines set by the manufacturer. This test enables the detection of protein S-nitrosocysteine (CySNO) PTMs. Briefly, S-nitrosylated proteins were treated with MMTS to restrict free sulfhydryls, followed by protein precipitation with acetone to eliminate MMTS. Then, S-nitrosylated proteins were modified with biotin (0.2 mM biotin-HPDP, Thermo Scientific, Waltham, MA, USA) in HPDP buffer for 1h at 4 °C in the dark. Next, biotinylated proteins were purified using acetone followed by pulling down in neutralizing buffer with streptavidin-agarose (GE healthcare, Waukesha, WI, USA). The beads were washed in neutralizing buffer with 600 mM NaCl, and then samples were eluted with an elution buffer containing 20 mM HEPES, 100 mM NaCl, 1 mM EDTA, 100 mM 2-ME. Western blot was employed to detect were detected by chemiluminescence (Bio-Rad, Hercules, CA, USA). The proteins of interest were measured with relevant antibodies. 

### 2.11. Data Analysis 

Data are shown as mean ± SEM. To compare multiple groups, one-way ANOVA with Tukey post hoc was applied. All statistical analyses were performed with GraphPad Prism version 8.0 (GraphPad Software, San Diego, CA, USA). A level of significance was set at *p* ≤ 0.05.

## 3. Results

### 3.1. IBF-R Controls Body Weight Gain and Its Influence on Metabolic Profile in HFD Induced Obese Mice

First, C57BL/6J mice were fed with 60% HFD for 12 weeks with or without IBF-R supplementation. IBF-R supplementation regulated the weight gain than the HFD fed mice (Figure 1a,b). Further, a micro-CT scan showed higher visceral adipose tissue in the HFD group than in the NCD group. Contrastingly, HFD fed mice with IBF-R supplementation dose-dependently reduced fat accumulation (Figure 1c). This observation was verified with the measurement of volume with the micro-CT program, where it indicated dose-dependent lower orbital fat volume (%) with IBF-R treatment (Figure 1d). Similar observations were recorded with respect to the weight of the liver, eWAT, and iWAT (Figure 1e–g). Interestingly, food intake in all the groups remained the same (Appendix A).

### 3.2. IBF-R Controls Biochemical Characteristics and Ameliorates Hepatic Lipid Accumulation in HFD Induced Obese Mice

Lipid contents were scrutinized to evaluate the influence of IBF-R on lipid homeostasis. NCD and HFD mice without IBF-R treatment showed considerably higher serum TG and TC levels, while IBF-R demonstrated significantly lower TG and TC levels (Figure 2a,b). IBF-R played a significant role in restoring blood glucose levels (Appendix A). Moreover, IBF-R supplementation improved adiponectin and lowered leptin levels dose-dependently (Figure 2c,d). Histological analysis of the liver revealed increased deposition of lipid droplets in HFD fed mice while IBF-R treatment drastically cut down the accumulation of lipid droplets (Figure 2e). 

### 3.3. IBF-R Regulates Adipogenesis through Redox-Mediated Post-Translational Modifications (PTMs) of IRE1α and ER Stress Response in HFD Induced Obese Mice

IRE1α is involved in ER homeostasis by causing a translational frameshift by starting unusual splicing of the mRNA encoding X-box-binding protein 1 (XBP1). It creates spliced XBP1 (sXBP1), a strong transcription factor that controls the expression of genes that encode ER chaperones [4] and proteins influencing phospholipid synthesis and afresh lipogenesis [17]. We observed enhanced IRE1α phosphorylation, sXBP-1, c-Jun N-terminal kinase (JNK) activation, GRP78, and CHOP expression, indicating preserved activation of the canonical ER stress sensors (Figure 3a,b). In response to ER stress, ER stress axis-based UPR is generated where IRE1α activates XBP-1 via unusual splicing of XBP-1 mRNA, followed by translocation of sXBP1 into the nucleus for the induction of chaperone proteins restoring ER homeostasis [18]. In Figure 3c, the IRE1α executor signaling “the nuclear translocation of sXBP-1” was slightly observed in HFD group. In contrast, the reduced translocation was significantly recovered in the IBF-R-treated groups (quantified in Figure 3d), suggesting a distinct molecular mechanism against metabolic disorders and obesity than recognized ER stress axis “IRE1α phosphorylation-sXBP-1 activation. 

Hence, we examined the ER quality control state through ER quality-control system [19]. Disruptions in the folding of client proteins do not allow it to attain mature conformation such as HMWC [20]. These multiprotein complexes were enhanced in HFD conditions (Figure 3e), which IBF-R attenuated. In obesity showing ER stress conditions, we asserted that obesity-induced and phosphorylation-independent modifications in IRE1α might suppress ribonuclease activity selectively. We observed a substantial rise in iNOS expression in HFD conditions, which coincides with the decrease in the sXBP1 activity “the nuclear translocation of sXBP-1” in adipose tissue of obese mice (Figure 3f) [16]. This coincidence represents an obesity model featuring chronic metabolic inflammation [21,22,23], whereas IBF-R significantly suppressed the iNOS level. Besides, other NO enzymes and endothelial NOS (eNOS) expression levels were comparable in the NCD and HFD conditions. S-nitrosylation, a protein modification involving covalent binding of nitrogen monoxide group to the thiol side chain of cysteine residues, appeared to be the underlying molecular process for active posttranslational regulation of proteins such as PDI [24]. In this study, we evaluated general protein S-nitrosylation (SNO) changes in the adipose tissues of NCD and HFD mice using biotin switch assay [25,26]. Protein S-nitrosylation was greatly increased in the HFD condition but significantly reduced in the IBF-R-treated conditions. (Figure 3g). Next, S-nitrosylated proteins were extracted from the adipose tissues of HFD and NCD mice to evaluate the influence of varying nitrosylation status affecting ER function. In addition, proteins critical to ER stress and adaptive responses were detected. These evaluations showed that multiple ER chaperones such as IRE1α and PDI demonstrated enhanced S-nitrosylation (Figure 3h) in HFD conditions while IBF-R decreased S-nitrosylation. These findings show that S-nitrosylation targets IRE1α raising the potential changes in the functioning of IRE1α in obese adipose tissue. 

### 3.4. IBF-R Controls Adipogenesis-Linked Proteins in Adipose Tissues in HFD Induced Obese Mice

H&E staining demonstrated that the IBF-R-treated condition is highly efficient in reducing the size and diameter of the adipocytes than the HFD group (Figure 4a,b). Later, the expressions of adipogenic transcription factors in eWAT were analyzed to clarify the contribution of reduced epididymal WAT (eWAT) in IBF-R-treated mice. Moreover, IBF-R significantly lowered the protein expressions of C/EBPα, SREBP1, PPARγ, and FAS than in the HFD group (Figure 4c–f). Lipid accumulation in adipose cells is due to enhanced lipogenesis and reduced β-oxidation [22]. Furthermore, markers of fatty acid oxidation and lipogenesis in eWAT and liver were evaluated to determine the molecular mechanisms of lowering eWAT mass upon IBF-R supplementation. Moreover, we scrutinized the expression of adipogenic transcription factors in eWAT to understand the process responsible for eWAT mass reduction upon IBF-R administration. The supplementation of IBF-R showed reduced SREBP1, PPARγ, C/EBPα, and FAS levels than the HFD group (Figure 4g–k). Similarly to eWAT, in liver tissues, HFD increased the adipogenic protein expression of SREBP1, PPARγ, C/EBPα, and FAS compared to the NCD group, whereas IBF-R administration significantly downregulated SREBP1, PPARγ, C/EBPα, and FAS compared to the HFD group (Appendix A).

In addition, p-AMPK was relatively less expressed in the HFD group than in the NCD group, while IBF-R significantly enhanced p-AMPK expression (Figure 4l,m). Next, SIRT1 expression was then examined to see if it was involved in regulating genes linked to adipogenic, lipogenic, and fatty acid oxidation. Interestingly, SIRT1 was greatly expressed in the IBF-R group than in the HFD group (Figure 4l,m). Together, these observations suggest that IBF-R controls the weight of eWAT and liver via modifications in adipogenesis-associated transcription factors.

## 4. Discussion

This study revealed the potential influence of IBF-R on obesity and extrapolated mechanisms behind positive implications on the adipose tissue of HFD mice. IBF-R treatment inhibited HFD-induced iNOS production as well as elevation of IRE1α S-nitrosylation, a master ER stress response signal, and associated fat accumulation and adipogenesis. Our findings suggest the IRE1α S-nitrosylation-inflammation axis as a novel mechanism to explain how IBF-R treatment regulates obesity. In addition, the study reveals that AMPK-SIRT1 activation is an IBF-R dependent mechanism in the obesity model.

IBF-R regulated body weight increase and its associated lipid dysmetabolism with which adipocyte differentiation is related. Obesity is characterized by excessive fat deposition that heavily impacts weight gain. IBF-R treatment significantly controlled the increase in body weight and fat accumulation without affecting food consumption. Furthermore, total cholesterol, triglyceride, and LDL-chol levels were considerably lower in the IBF-R treatment group. Several transcriptional factors such as PPARγ and C/EBP family proteins mediate adipogenesis [27]. The expressions of C/EBPβ and C/EBPδ stimulates the expressions of C/EBPα and PPARγ, which are considered critical positive regulators of adipogenesis during the early stages of adipocyte development. In addition, PPARγ is the key regulator during adipocyte development and differentiation [28]. Cells with PPARγ deficiency halt the development of adipocytic cells. As a result, they could not develop as matured adipocytes although unconventionally expressed of other pro-adipogenic factors [29]. In this investigation, IBF-R supplementation exhibited downregulation of PPARγ and C/EBP along with its downstream FAS than the HFD condition. Other transcriptional factors involved in adipogenesis modulation include CREB1 and SREBP1c. Specifically, SREBP1c is required for adipocyte differentiation and can stimulate PPAR and FAS expression [30]. However, IBF-R significantly reduced SREBP1c and FAS in the HFD model. These observations back up the concept that IBF-R treatment reduced adipogenesis in eWAT of HFD induced obese mice. 

IBF-R regulated IRE1α post-translational modifications (PTMs), leading to ER homeostasis and its related lipid homeostasis. S-nitrosylation of IRE1α suppresses its endoribonuclease activity, preventing sXBP1 synthesis, thereby preventing downstream targets such as decreased ER chaperones, malfunctioned autophagy or ERAD failure, all of which are required to preserve ER homeostasis. Obesity is related to increased iNOS-induced metaflammation and reduced XBP1 processing [31], similar to study findings in the HFD-induced obesity model (Figure 3c). Obesity, insulin resistance (IR), and diabetic nephropathy exhibit faulty nuclear translocation of sXBP1 [32,33,34], where the link within sXBP1 and PI3K regulatory subunits (p85α and p85β) was disrupted, exacerbating disease etiology [34,35]. Study data suggest that IBF-R passes IRE1α-mediated XBP1 splicing, relying on redox-mediated PTMs of IRE1α into SNO, facilitating modifications of IRE1α. Contentious stimulation of SNO decides whether IRE1 RNase splicing activity is activated or inhibited, giving critical signals that decide cell adaptation or death. IBF-R demonstrated its potentials in regulating adipogenesis by sustaining redox-linked PTMs and ER homeostasis. Moreover, previously, we have demonstrated that severe redox-mediated PTMs of IRE1α impairs ER function and sustain ER stress in aging/metabolic disorder models suggesting the S-nitrosylation of IRE1α affects its endoribonuclease activity, thereby impairs sXBP1, leading to inflammation [16]. In addition, the increased iNOS-induced metaflammation is associated with impaired XBP1 processing in obesity [31]. A defective nuclear translocation of sXBP1 has been reported in metabolic disorders such as obesity and diabetes [32,33], indicating the potential use of IRE1α modifications, i.e., the IBF-R might prevent or control obesity.

The other finding is that IBF-R enhanced the AMPK-SIRT1 axis controlling adipogenesis in HFD model. In addition, we observed IBF-R stimulated AMPK expression influencing fatty acid metabolism and adipose tissue development (Figure 4l). Additionally, AMPK activation is linked to a reduction in lipid accumulation [36]. Furthermore, it also inhibits FAS and alleviates fatty acid oxidation by phosphorylating ACC [37]. A769662, an activator of AMPK, reduces lipid droplets, PPARγ, and C/EBP [38]. Due to obesity, AMPK becomes inactive necessitating external stimuli to activate AMPK. SIRT1 is a key member of the sirtuin family that plays a vital role in improving the fighting capacity of the cell against stress in multiple ways, including regulation of aging, metabolic activity and apoptosis, and ER stress [39]. SIRT1 deacetylates XBP1 blocking its transcriptional activity, enhancing ER stress-linked programmed cell death, and decreasing translational inhibition dependent on PERK-eIF2 in mice [40,41]. Hence, SIRT1 is considered a negative regulator of the ER stress response. Here, IBF-R was observed to be regaining SIRT1, p-AMPK, which controls SREBP-1c signaling and the associated ER stress response (Figure 4l,m). These findings suggest that IBF-R protects against obesity via reduced adipocyte formation and lipogenesis in the eWAT, potentially with enhanced AMPK/SIRT1 activation linked with ER stress.

In summary, the study revealed that supplementation of IBF-R cut down the risks of obesity in the HFD model via repressing fat accumulation in the WAT. Concerning molecular mechanisms, treatment with IBF-R upholds AMPK-SIRT1 signaling and modulates SREBP1-ER stress. Collectively, investigation adds strong evidence on the health benefits of RV extract and lays a firm foundation for the development of RV extract as a functional ingredient in the food and pharmaceutical industry.

## Figures and Tables

**Figure 1 nutrients-14-00217-f001:**
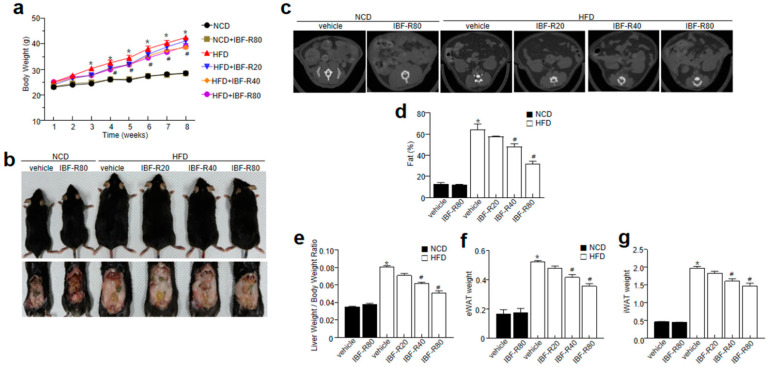
Influence of IBF-R on body weight and fat mass. Mice were fed with NCD or HFD with vehicle or IBF-R (20, 40, or 80 mg/kg). All the animals were gavaged daily for 12 weeks. (**a**) Variations in body weight at predetermined time points. (**b**) Representative images of HFD mice after 12 weeks of IBF-R supplementation. (**c**) Representative Micro-CT images were obtained using Skyscan1076 micro-CT scanner. (**d**) Quantification of fat mass with Skyscan1076 micro-CT scanner at the end of the experiment. (**e**–**g**) Measurement of epididymal white adipose tissue (eWAT), inguinal white adipose tissue (iWAT) and liver weight. Data are shown as mean ± SEM (*n* = 10, * *p* < 0.05 vs. NCD + vehicle, # *p* < 0.05 vs. HFD + vehicle).

**Figure 2 nutrients-14-00217-f002:**
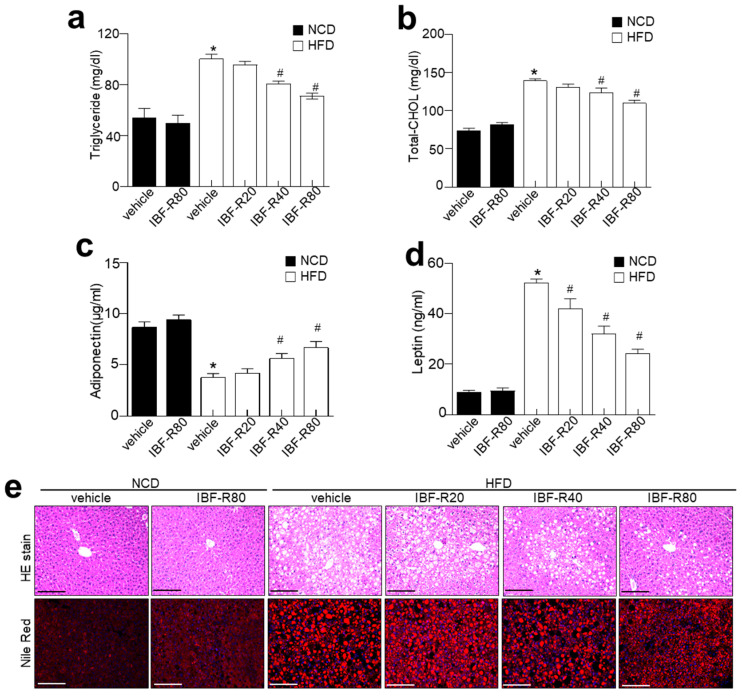
Influence of IBF-R serum biochemicals profile and lipid accumulation. Levels of triglyceride (**a**), total cholesterol (**b**), adiponectin (**c**), and leptin (**d**). (**e**) Representative images of liver sections stained with H&E (upper) and Nile Red stain (lower). Scale bars = 50 µm. Data are shown as mean ± SEM (*n* = 10, * *p* < 0.05 vs. NCD + vehicle, # *p* < 0.05 vs. HFD + vehicle).

**Figure 3 nutrients-14-00217-f003:**
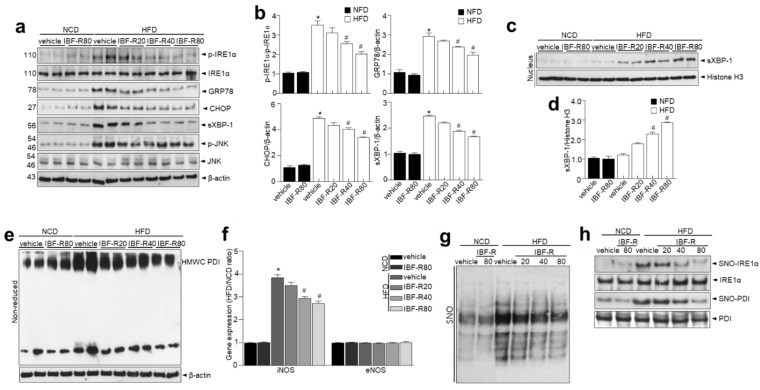
IBF-R diminish ER stress and IRE1α S-nitrosylation of axis in HFD model. (**a**) Immunoblotting of p-IRE1α, IRE1α, GRP78, CHOP, sXBP-1, and β-actin expressions in eWAT. (**b**) Quantitative analysis of proteins. (**c**,**d**) sXBP1 expressions at the nuclear level from each group and quantitative analysis. (**e**) eWAT lysate was examined for PDI in HMWCs on non-reducing gels. (**f**) iNOS and eNOS mRNAs were measured in eWAT by qRT-PCR. (**g**) General S-nitrosylation (SNO) profile in eWAT of HFD mice and NCD controls. (**h**) Specific SNO proteins in the eWAT of HFD mice and NCD controls. S-nitrosylated proteins were purified with biotin-switch assay and detected by immunoblotting. Data are shown as mean ± SEM (*n* = 10, * *p* < 0.05 vs. NCD + vehicle, # *p* < 0.05 vs. HFD + vehicle). eWAT; epididymal white adipose tissue.

**Figure 4 nutrients-14-00217-f004:**
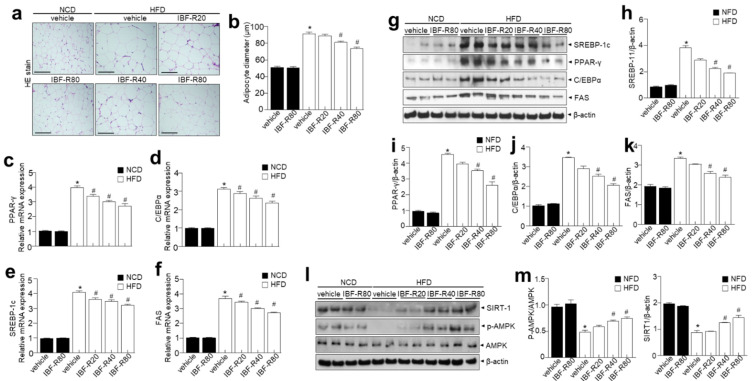
IBF-R determines adipose tissue expansion and adipogenic factors in eWAT. (**a**) H&E staining in eWAT. Scale bar = 50 µm. (**b**) The mean diameter of adipocytes in eWAT. (**c**–**f**) C/EBPα, PPAR-γ, FAS and SREBP-1c were measured in eWAT by qRT-PCR. (**g**) Immunoblotting of PPAR-γ, C/EBPα, SREBP-1c, FAS, and β-actin expressions in eWAT. (**h**–**k**) Quantification of protein expressions. (**l**,**m**) Immunoblotting of p-AMPK, AMPK, SIRT-1, and β-actin expressions in eWAT and respective quantitative analysis. Data are shown as mean ± SEM (*n* = 10, * *p* < 0.05 vs. NCD + vehicle, # *p* < 0.05 vs. HFD + vehicle). eWAT; epididymal white adipose tissue.

## Data Availability

The data presented in this study are available upon request from the corresponding author.

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
