# Peer review of "IBF-R Regulates IRE1α Post-Translational Modifications and ER Stress in High-Fat Diet-Induced Obese Mice"

_nutrients, 2022, doi:10.3390/nu14010217_

Round 1
Reviewer 1 Report
The author investigated the anti-obesity effect RV extract, a traditional food supplement, in HFD-mice. And author also find that RV extract diminished ER stress and PTM of IRE1-alpha in HFD mice, which explained the mechanism of RV extract in prevent against obesity. It is a worthwhile finding as author provided some strong evidences that RV extract did attenuated the process of obesity in HFD-mice. Unfortunately, the mechanism of action of EV extract is not clearly demonstrated. Section 3.3 is not well organized, though the data are of good quality, but the presentation and verbal description are obscure, the conclusion seems to be indiscreet. RV-mediated post-translational modifications (PTMs) of IRE1α and ER stress response may only be a plausible mechanism.
- It is not clear how IBF-R is prepared and what is main components of IBF-R. Is IBF-R a combination of Fisetin and Fustin? It is confusing why author mentioned “Fisetin and fustin is a major flavonol compound found in IBF-R extract”(P4, line 166). Please demonstrate detailed method of preparation of IBF-R.
- It is not clear explained the significance of nuclear translocation of sXBP1 (Fig 3c).
- The sentence needs to be reorganized (P6, Line 212-214).
- the author did not mention which tissue is subjected to analysis in result (Figure3).
- It is not clear the up-regulation of S-nitrosylation of IRE1-alpha is the cause or effect of HFD-induced obesity, and attenuation of ER-stress and PTM of IRE1 is indeed account for the mechanism of anti-obesity effect of IBF-R. More strong evidences are required. Please give more information regarding the relation between ER stress and PTM of IRE1 and adipogenesis.
Author Response
Reviewer 1
The author investigated the anti-obesity effect RV extract, a traditional food supplement, in HFD-mice. And author also find that RV extract diminished ER stress and PTM of IRE1-alpha in HFD mice, which explained the mechanism of RV extract in prevent against obesity. It is a worthwhile finding as author provided some strong evidences that RV extract did attenuated the process of obesity in HFD-mice. Unfortunately, the mechanism of action of EV extract is not clearly demonstrated. Section 3.3 is not well organized, though the data are of good quality, but the presentation and verbal description are obscure, the conclusion seems to be indiscreet. RV-mediated post-translational modifications (PTMs) of IRE1α and ER stress response may only be a plausible mechanism.
Q1. It is not clear how IBF-R is prepared and what is main components of IBF-R. Is IBF-R a combination of Fisetin and Fustin? It is confusing why author mentioned “Fisetin and fustin is a major flavonol compound found in IBF-R extract”(P4, line 166). Please demonstrate detailed method of preparation of IBF-R.
A1. We apologize for the inconvenience about inappropriate description “Fisetin and fustin is a major flavonol compound found in IBF-R extract”. In this study, we have never studied about the main components but about the Rhus verniciflua water extracts. The extract has a lot of main components including fistein and fustin. However, in our animal study, the pure components were not applied due to the low bioavailability. Instead, the Rhus verniciflua water extract, a popular formulation in Eastern Asia, was applied to the dyslipidemia model, in vivo.
Following the comment, we detailed the preparation of IBF-R protocol in the revised version as follows.
“2.1. The preparation of IBF-R
- vernicifluawas obtained from the Imsil Cheese & Food Research Institute (Imsil-gun, Jeollabuk-do, Korea). The R. vernicifluawas processed as described before [1]. Briefly, R. verniciflua was finely pulverized, hot air dried, extracted with boiling distilled water, concentrated under reduced pressure using a rotary evaporator, and lyophilized to obtain dried R. verniciflua extract (IBF-R). The yield of the lyophilized R. verniciflua extract was 4%. Finally, the botanical extract of R. verniciflua is given the name IBF-R. Briefly, R. verniciflua (1,000 g) was extracted with 5,000 mL of distilled water for 2 h at 121 °C. The R. verniciflua extracts were centrifuged at 5,000g for 20 min at 4 °C (Himac CR-22F, Hitachi Koki Co., Ltd., Tokyo, Japan), and the supernatants were filtered using filter paper (Whatman No.1, Sigma-Aldrich, St. Louis, USA). The filtrate was concentrated with a rotary evaporator and then lyophilized with a freeze dryer (Ilshin Lab Co. Ltd., Seoul, Korea). The quality control was performed using standard compounds such as Fisetin as described before [2].”
Q2. It is not clear explained the significance of nuclear translocation of sXBP1 (Fig 3c).
A2. Following the comment, we updated the explanation of the significance of nuclear translocation of sXBP1 in the revised version as follows.
“In response to ER stress, IRE1α activates XBP-1 through unconventional splicing of XBP-1 mRNA, followed by translocation of spliced XBP-1 (sXBP1) into the nucleus for the induction of chaperone proteins which restore ER homeostasis [3]; an ER stress axis-based unfolded protein response. In Figure 3c, the IRE1α executor signaling “the nuclear translocation of sXBP-1” was slightly observed in HFD group. In contrast, the reduced translocation was significantly recovered in the IBF-R-treated groups (quantified in Figure 3d), suggesting a distinct molecular mechanism against metabolic disorders and obesity than recognized ER stress axis "IRE-1α phosphorylation-sXBP-1 activation."
Q3. The sentence needs to be reorganized (P6, Line 212-214).
A3. Following the comment, we reorganized in Result section in revised version as follows.
“In Figure 3c, the IRE1α executor signaling “the nuclear translocation of sXBP-1” was only slightly observed in HFD group. In contrast, the reduced translocation was significantly recovered in the IBF-R-treated groups (quantified in Figure 3d), suggesting a distinct molecular mechanism against metabolic disorders and obesity than recognized ER stress axis "IRE1α phosphorylation-sXBP-1 activation.”
Q4. the author did not mention which tissue is subjected to analysis in result Figure3.
A4. Following the comment, we have updated the Figure Legends in the revised version as follows.
Figure 3. IBF-R diminish ER stress and IRE1α S-nitrosylation of axis in HFD model. (a) Immunoblotting of p-IRE1α, IRE1α, GRP78, CHOP, sXBP-1, and β-actin expressions in eWAT. (b) Quantitative analysis of proteins. (c-d) The expression of sXBP1 shown at the nuclear level from each group and quantitative analysis. (e) eWAT lysate was analyzed for the presence of PDI in HMWCs on non-reducing. (f) iNOS and eNOS mRNAs were measured in eWAT by qRT-PCR. (g) General SNO profile in eWAT of HFD mice and NCD controls. Purification of S-nitrosylated proteins with biotin-switch assay and then identified via Western blot analysis. (h) Specific SNO proteins in the eWAT of HFD mice and NCD controls. The S-nitrosylated proteins were purified by a biotin-switch method and detected by Western blot analysis. Data are presented as mean ± SEM (n = 10, *p < 0.05 vs NCD + vehicle, #p < 0.05 vs HFD + vehicle). eWAT; epididymal WAT”
“Figure 4. IBF-R determines adipose tissue expansion and adipogenic factors in eWAT. (a) H&E staining in eWAT. Scale bar=50 µm (b) The average diameter of adipocytes in eWAT. (c-f) PPAR-γ, C/EBPα, SREBP-1c, and FAS were measured in eWAT by qRT-PCR. (g) Immunoblotting of SREBP-1c, PPAR-γ, C/EBPα, FAS, and β-actin expressions in eWAT. (h-k) Quantitative analysis of protein expression was also performed. (l-m) Immunoblotting of SIRT-1, p-AMPK, AMPK, and β-actin expressions in eWAT and respective quantitative analysis. Data are presented as mean ± SEM (n = 10, *p < 0.05 vs NCD + vehicle, #p < 0.05 vs HFD + vehicle). eWAT; epididymal WAT”
Q5. It is not clear the up-regulation of S-nitrosylation of IRE1-alpha is the cause or effect of HFD-induced obesity, and attenuation of ER-stress and PTM of IRE1 is indeed account for the mechanism of anti-obesity effect of IBF-R. More strong evidences are required. Please give more information regarding the relation between ER stress and PTM of IRE1 and adipogenesis.
A5. Following the comment, we updated the discussion part in the revised version.
“IBF-R regulated IRE1α PTM leading to ER homeostasis and its related lipid homeostasis. S-nitrosylation of IRE1α suppresses its endoribonuclease activity, preventing sXBP1 synthesis or nuclear translocation, thereby preventing downstream targets, such as decreased ER chaperones, ERAD failure, or malfunctioned autophagy, all of which are required for maintaining ER homeostasis. Obesity is related to increased iNOS-induced metaflammation and reduced XBP1 processing [21], similar to study findings in the HFD-induced obesity model (Fig. 3c). Obesity, insulin resistance, and diabetic nephropathy exhibit faulty nuclear translocation of sXBP1 [51-53], where interaction between sXBP1 and regulatory subunits of PI3K (p85α and p85β) was disrupted, exacerbating disease etiology [4, 51]. Study data suggest that IBF-R passes IRE1α-mediated XBP1 splicing, relying on redox-mediated post-translational modifications (PTMs) of IRE1α into SNO, which facilitate the alterations in IRE1α function. Competitive stimulation of SNO decides whether IRE1 RNase splicing activity is activated or inhibited, giving critical signals that decide cell adaptation or death. IBF-R demonstrated its potentials in regulating adipogenesis by sustaining redox-linked PTMs and ER homeostasis. Our group has also demonstrated that severe redox-mediated PTMs of IRE1α impairs ER function and prolongs ER stress in human and mice aging/metabolic disorder models suggesting that S-nitrosylation of IRE1α affects its endoribonuclease activity, thereby impairs sXBP1, leading to inflammation [4]. In addition, the increased iNOS-induced metaflammation is associated with impaired XBP1 processing in obesity [5]. A defective nuclear translocation of sXBP1 has been reported in metabolic disorders such as obesity and diabetes [6,7], indicating that the inhibition strategy of IRE1α modifications, i.e., the IBF-R might prevent or control obesity.”
References
- Lee, H.Y.; Lee, G.H.; Yoon, Y.; Chae, H.J. R. verniciflua and E. ulmoides Extract (ILF-RE) Protects against Chronic CCl(4)-Induced Liver Damage by Enhancing Antioxidation. Nutrients 2019, 11, doi:10.3390/nu11020382.
- Hoang, T.H.; Yoon, Y.; Park, S.A.; Lee, H.Y.; Peng, C.; Kim, J.H.; Lee, G.H.; Chae, H.J. IBF-R, a botanical extract of Rhus verniciflua controls obesity in which AMPK-SIRT1 axis and ROS regulatory mechanism are involved in mice. J Funct Foods 2021, 87, doi:ARTN 10480410.1016/j.jff.2021.104804.
- Calfon, M.; Zeng, H.; Urano, F.; Till, J.H.; Hubbard, S.R.; Harding, H.P.; Clark, S.G.; Ron, D. IRE1 couples endoplasmic reticulum load to secretory capacity by processing the XBP-1 mRNA. Nature 2002, 415, 92-96, doi:10.1038/415092a.
- Bhattarai, K.R.; Kim, H.K.; Chaudhary, M.; Ur Rashid, M.M.; Kim, J.; Kim, H.R.; Chae, H.J. TMBIM6 regulates redox-associated posttranslational modifications of IRE1alpha and ER stress response failure in aging mice and humans. Redox Biol 2021, 47, 102128, doi:10.1016/j.redox.2021.102128.
- Yang, L.; Calay, E.S.; Fan, J.; Arduini, A.; Kunz, R.C.; Gygi, S.P.; Yalcin, A.; Fu, S.; Hotamisligil, G.S. METABOLISM. S-Nitrosylation links obesity-associated inflammation to endoplasmic reticulum dysfunction. Science 2015, 349, 500-506, doi:10.1126/science.aaa0079.
- Madhusudhan, T.; Wang, H.; Dong, W.; Ghosh, S.; Bock, F.; Thangapandi, V.R.; Ranjan, S.; Wolter, J.; Kohli, S.; Shahzad, K., et al. Defective podocyte insulin signalling through p85-XBP1 promotes ATF6-dependent maladaptive ER-stress response in diabetic nephropathy. Nat Commun 2015, 6, 6496, doi:10.1038/ncomms7496.
- Park, S.W.; Zhou, Y.; Lee, J.; Lu, A.; Sun, C.; Chung, J.; Ueki, K.; Ozcan, U. The regulatory subunits of PI3K, p85alpha and p85beta, interact with XBP-1 and increase its nuclear translocation. Nat Med 2010, 16, 429-437, doi:10.1038/nm.2099.

Reviewer 2 Report
Reviewers
General comment-This is a clearly presented and well-written paper. In this manuscript, the authors examined the anti-obesity effects of IBF-R from Rhus verniciflura (RV) extract in high fat diet murine model. IBF-R regulated the body weight and fat contents. IBF-R supplementation significantly altered the lipid homeostasis by reducing the lipogenesis and improving fatty acid oxidation. Obesity is associated with metabolic inflammation, ER stress and stress induced PTM of IRE1alpha. Treatment with IBF-R ameliorates these pathological changes in diet induced obese mice. This study highlights the role of IBF-R on obesity and explains the IRE1a S-nitrosylation-inflammation axis as a possible mechanism of pathogenesis in obesity.
Summary of the salient findings:
In high fat diet induced mice, treatment with IBF-R reduced body weight, fat mass and corrects hepatic lipid dysregulation. IBF-R diminishes ER stress and IRE1a S-nitrosylation. Also, activates AMPK and SIRT1 in adipose tissue of high fat diet model.
The proposed study is interesting, but I have the following comments and concerns.
- Study revealed that IBF-R lowered the adipogenic transcription factors such as PPAR. What about status of other transcription factors which are critical for regulation of lipid homeostasis along with PPARs such as LXR and FXR and their target genes especially in liver?
- This study explains in detail the role of IBF-R in obese mice model, will it be effective in clinical settings? Any explanation.
- In methods- Biochemical analysis section not explained in detail and WB analysis section- Same sentence repeated twice in the beginning.
- Last authors name looks missing in authors list (list ends with and3).
Author Response
Reviewers 2
General comment-This is a clearly presented and well-written paper. In this manuscript, the authors examined the anti-obesity effects of IBF-R from Rhus verniciflura (RV) extract in high fat diet murine model. IBF-R regulated the body weight and fat contents. IBF-R supplementation significantly altered the lipid homeostasis by reducing the lipogenesis and improving fatty acid oxidation. Obesity is associated with metabolic inflammation, ER stress and stress induced PTM of IRE1alpha. Treatment with IBF-R ameliorates these pathological changes in diet induced obese mice. This study highlights the role of IBF-R on obesity and explains the IRE1a S-nitrosylation-inflammation axis as a possible mechanism of pathogenesis in obesity.
Summary of the salient findings:
In high fat diet induced mice, treatment with IBF-R reduced body weight, fat mass and corrects hepatic lipid dysregulation. IBF-R diminishes ER stress and IRE1a S-nitrosylation. Also, activates AMPK and SIRT1 in adipose tissue of high fat diet model.
The proposed study is interesting, but I have the following comments and concerns.
Q1. Study revealed that IBF-R lowered the adipogenic transcription factors such as PPAR. What about status of other transcription factors which are critical for regulation of lipid homeostasis along with PPARs such as LXR and FXR and their target genes especially in liver?
A1. Following the comment, we updated the SREBP1, PPAR-γ, CEBP1, and FAS in liver in the revised version (Figure S2).
We updated the Result part in the revised version as follows.
“Similar to eWAT, HFD increased the adipogenic protein expression of SREBP1, PPARγ, C/EBPα, and FAS compared to the NCD group, whereas IBF-R administration significantly downregulated SREBP1, PPARγ, C/EBPα, and FAS compared to the HFD group (Figure S2).”
Q2. This study explains in detail the role of IBF-R in obese mice model, will it be effective in clinical settings? Any explanation.
A2. We appreciate for the important comment. Rhus verniciflua has been well known to contribute to anti-obesity and anti-inflammatory effect. However due to the toxic component, Urushiol, Rhus verniciflua was not allowed to be transferred to Clinical Trials. When we overcome the hurdle “the perfect removal of Urushiol”, many clinical trials would be performed, it is expected.
We believe that we need to accumulate the beneficial effect of Rhus verniciflua extract against metabolic disorder although in vivo animal experiment. In the routinely applied doses, there is no toxic effect at all. In the extremely high doses, there can be a toxic effect, a kind of task to develop the extract as a functional food. In the near future, another formulations of Rhus verniciflua extract would be documented in human-based clinical studies. For the readability, we did not mention the point in the revised version.
Q3. In methods- Biochemical analysis section not explained in detail and WB analysis section- Same sentence repeated twice in the beginning.
A3. Following the comment, we updated the information about Biochemical analysis section in the revised Methods as follows.
“2.2. Biochemical Analysis
Biochemical analysis was performed using commercial kits. Blood was placed in tubes containing EDTA2Na, and serum was obtained by centrifuging the blood at 3000×g 10 min at 4 °C. Serum levels of triglyceride (TG, AM1575K) and total-cholesterol (AM202K) were detected using a using commercial kits according to manufacturer’s guidelines (Asan phams, Co., Korea). Plasma levels of adiponectin and leptin were determined using commercial sandwich ELISA kits (CSB-E07272m, CSB-E04650m, CUSABIO, USA). The absorbance at 450 nm was detected using a microplate reader (Multiskan SkyHigh; Thermo Fisher Scientific, Inc.). The unknown sample concentration was determined from the standard curve, and concentration values were reported in ng/mL.”
Additionally, we deleted the repeated sentence in the Western blotting section in the revised version.
“2.4. Western blot analysis
Western blotting was carried out as suggested previously [1]. All the immunoblots were probed with relevant antibodies. Antibodies used in this study are AMP-activated kinase (AMPK, #2532) and phosphorylation of AMP-activated kinase (p-AMPK, P-2535) (Cell Signaling Technologies, Inc., Danvers, MA, USA) and sterol regulatory element-binding protein (SREBP-1c, sc-36553), peroxisome proliferator-activated receptor γ (PPAR-γ), CCAAT/enhancer-binding protein α (C/EBP1α, sc-166258), fatty acid synthase (FAS, sc-74540), sirtuin 1 (SIRT1, sc-74465), β-actin (sc-47778, Santa Cruz, Biotechnologies, Inc., Santa Cruz, CA, USA).”
Q4. Last authors name looks missing in authors list (list ends with and3).
A4. We apologize for the mistake. We corrected the author list.
References
- Hoang, T.-H.; Yoon, Y.; Park, S.-A.; Lee, H.-Y.; Peng, C.; Kim, J.-H.; Lee, G.-H.; Chae, H.-J. IBF-R, a botanical extract of Rhus verniciflua controls obesity in which AMPK-SIRT1 axis and ROS regulatory mechanism are involved in mice. Journal of Functional Foods 2021, 87, 104804, doi:https://doi.org/10.1016/j.jff.2021.104804.
